# Identifying Communities in Dynamic Networks Using Information Dynamics

**DOI:** 10.3390/e22040425

**Published:** 2020-04-09

**Authors:** Zejun Sun, Jinfang Sheng, Bin Wang, Aman Ullah, FaizaRiaz Khawaja

**Affiliations:** 1School of Computer Science and Engineering, Central South University, Changsha 401302, China; szj@pdsu.edu.cn (Z.S.); dr.aman@csu.edu.cn (A.U.); riazfaiza94@yahoo.com (F.K.); 2School of Information Engineering, Pingdingshan University, Pingdingshan 462500, China

**Keywords:** dynamic community detection, information dynamics, propagation, cluster

## Abstract

Identifying communities in dynamic networks is essential for exploring the latent network structures, understanding network functions, predicting network evolution, and discovering abnormal network events. Many dynamic community detection methods have been proposed from different viewpoints. However, identifying the community structure in dynamic networks is very challenging due to the difficulty of parameter tuning, high time complexity and detection accuracy decreasing as time slices increase. In this paper, we present a dynamic community detection framework based on information dynamics and develop a dynamic community detection algorithm called DCDID (dynamic community detection based on information dynamics), which uses a batch processing technique to incrementally uncover communities in dynamic networks. DCDID employs the information dynamics model to simulate the exchange of information among nodes and aims to improve the efficiency of community detection by filtering out the unchanged subgraph. To illustrate the effectiveness of DCDID, we extensively test it on synthetic and real-world dynamic networks, and the results demonstrate that the DCDID algorithm is superior to the representative methods in relation to the quality of dynamic community detection.

## 1. Introduction

Systems in the real world can be abstracted into complex networks, and a large number of algorithms for complex network mining have been proposed [1,2,3,4,5], most of which focus on static networks. However, most networks in the real world are evolving over time. For example, news and public opinions are constantly being produced, spread and abandoned; infectious diseases are constantly appearing, spreading and dying; urban road networks are constantly being built, expanded and demolished; organizations or groups in social networks are constantly forming, growing and dissolving, and so on. Studying the structural characteristics of dynamic networks is beneficial to analyze the formation mechanism of networks, explore the implicit structure, and predict the evolution of the network structure. These have important theoretical and practical value for the development of various fields and disciplines in real society. For decades, researchers have conducted extensive research on community structures in complex networks and have proposed many well-known static community discovery methods, such as GN [6], CPM [7], LPA [8], Louvain [9], Infomap [10], MCL [11], etc. However, the network structure in the real world is not static. In contrast, the community structure of most networks is constantly evolving over time. The existing static community discovery methods divide the community based on the static topology of the network and have not considered the relationship between the network structures under multiple snapshots. Therefore, most static community detection algorithms are not suitable for dynamic networks.

Recently, community detection in dynamic networks has attracted increasing attention and become a popular research topic because of their high potential for apprehending social phenomena over time. Many methods have been designed for identifying the communities in dynamic networks from different perspectives, which include dynamic community detection based on evolutionary clustering [12] and incremental [13] dynamic community detection. The evolutionary clustering-based method defines a general clustering framework, so that some static clustering algorithms can also be extended to community detection in dynamic networks. Evolutionary clustering includes two evaluation indicators; one is the clustering quality of the current snapshot, and the other is the difference of the clustering result between the current and the previous snapshots. The methods based on evolutionary clustering consider both that the evolution of the network structure is smooth and the relationship between snapshots, but such methods still have a high time complexity. Similar to evolutionary clustering, the incremental dynamic community detection methods still assume that the network evolution is smooth. This type of algorithm first performs global community detection on the initial network, and then incrementally detects the changed subgraphs in the snapshot network. Therefore, incremental community detection methods often have faster detection speeds. Such methods have also attracted the attention of researchers and several incremental algorithms have been proposed [13,14,15]. However, most of the existing incremental dynamic community detection methods have gradually reduced their performance as the number of snapshots increases. In the process of incremental community identification, only the local structure of the network is detected; therefore, some changes in the associated structure may be neglected. In addition, when processing the addition and deletion of nodes and edges incrementally, the processing order may have an impact on the detection results and efficiency [16]. Therefore, accurately and efficiently identifying the community structure in a dynamic network is still a very interesting and challenging task.

In this paper, we propose a dynamic community detection framework based on information dynamics. Based on this framework, we design a dynamic community detection algorithm DCDID. We regard a network as a closed dynamic system with the information exchanges in the network based on the topology. Through the exchange of information between nodes, nodes with the same characteristics will self-organize, and finally, the community structure in the network will be naturally exposed. This new viewpoint provides a fresh approach to dynamic community detection and it has some attractive features. We will introduce DCDID in Section 3, but first, let us describe the basic idea of the proposed method.

### 1.1. Basic Idea

In social networks, a person usually prefers to communicate with people who share their values. This feature promotes the formation and development of groups, and groups also facilitate the communication of information between people. Therefore, the exchange of information among people plays a very important role in the formation and growth of communities. The structure of the network affects information propagation; in turn, the spread of information also reflects the community structure of the network. Based on these characteristics, we can construct models to automatically reveal the community structure by imitating the communication of information between people. Thus, we designed a new approach based on information dynamics to acquire insights into the division of communities in dynamic networks, where the basic idea is to consider an accommodative dynamical system and investigate its information dynamics over time. In particular, in an interpersonal network, people with similar interests or features are more likely to interact with others, and the propagation of information between them tends to be more frequent. With the diffusion and interaction of information, people in the same community have almost the equivalent amount of information, whereas those in diverse communities have different amounts of information. Over time, the information dynamics on the network reaches the steady state, and the communities can be naturally uncovered by counting the amount of information of nodes in the network.

To better illustrate the basic idea, we employ a toy network as an example. Figure 1 describes the process of information dynamics in a dynamic social network, which consists of many users denoted as cartoon persons with different colors. In this dynamic artificial network, we take a company as an example to present the process of dynamic community detection based on information dynamics, which comprises the following stages. First, everyone possesses his/her own knowledge as initial information because of distinct occupations (i.e., $v=0.66,u_{1}=0.5$), as shown in Figure 1a (see Definition 3). Then, the information spreads through the topological structure of the network (see Equation (Equation 4)). For example, user *v* shares its information with the connected neighbors $u_{1}-u_{4}$, as shown in Figure 1b. Over time, the amount of information exchanged between people tends to zero and the information dynamics reaches the steady state. Next, the communities are naturally revealed by computing the different information in the network. Figure 1b shows the community structure detected by the information dynamics model in the time slice $T_{0}$. Building upon the information of communities detected at the time slice $T_{0}$, an incremental community discovery framework is adopted for the subsequent snapshot networks (see Figure 2), which includes adding nodes, deleting nodes, adding edges, and deleting edges events. Figure 1c–f demonstrates that the addition and deletion of nodes and edges may lead to changes in the network structure. For example, the addition of user $u_{5}$ causes the two communities to merge into one community, as shown in Figure 1c. Similarly, Figure 1d,e present that the deletion of users $u_{6}$ and $u_{7}$ results in a split in the community.

### 1.2. Contributions

By imitating the information dynamics, the proposed dynamic community detection algorithm has several attractive benefits, the most important of which are listed as follows:**Effective Dynamic Community Detection**: We propose an information dynamics-based framework, which employs the batch processing technique to incrementally uncover community structures in dynamic networks. In addition, we develop a dynamic community detection method DCDID for revealing the communities by simulating the exchange of information between nodes in each time slice of dynamic networks. The DCDID method provides a natural manner to uncover community structure and obtains high-quality communities in each time slice (cf. Figures 7–12).**Parameter-free**: The DCDID approach does not require parameter settings and prior knowledge, and it automatically detects the communities through information dynamics driven by the local topological structure of the network.**Scalability**: Because of using the batch processing and incremental technology, DCDID only needs to update those communities where the subgraph changed and keep the rest of the subgraph unchanged in each time slice. Thus, DCDID has a low time complexity and can be applied to large-scale dynamic networks. (cf. Section 3.5).

The rest of this paper is organized as follows. In the second section, we introduce the related work. Section 3 gives the information dynamic model and presents the dynamic community detection framework and algorithm in detail. Section 4 describes the experimental results of DCDID on synthetic and real-world networks. Finally, our conclusions are presented in Section 5.

## 2. Related Work

In recent years, many methods have been proposed for community detection in dynamic networks. Here, we only summarize some contributions in the most closely related areas from the literature. For more detailed research work of dynamic community detection, please refer to the review literature in [17,18,19].

**Incremental-based Methods.** The topic of Incremental community detection in dynamic networks has attracted significant attention in recent years. Shang et al. [20] proposed a real-time detection algorithm in dynamic networks, which detects the community structure of the initial network based on the Louvain algorithm, and then employs the strategy of modularity optimization to update the community structure according to the type of the changed edge. Chong et al. [16] presented an incremental batch processing technique, which uses the Louvain method for batch processing of the changed structure. In addition, it pointed out some problems existing in the real-time processing algorithms based on event processing of nodes and edges, such as the impact of the processing sequence on detection results and efficiency. LBTR is an incremental community detection method based on machine learning, which was proposed by Shang et al. [14]. This method classifies and predicts nodes that need to be processed by machine learning methods, and filters unchanged nodes to improve the efficiency of community detection. Takaffoli et al. [21] introduced an Incremental L-metric community detection method, which employs the current and temporal data to identify the communities in each snapshot network.

**Evolution-based Methods.** Chakrabarti et al. [12] provided a general framework, which simultaneously optimizes two evaluation indicators: the clustering quality of each snapshot network and the difference of the clustering result from one snapshot to the next. Chi et al. [22] proposed an evolutionary spectral clustering that is an extension of evolutionary clustering. FacetNet [23] is a classic and widely used approach for analyzing communities and their evolutions in dynamic networks. It uses the nonnegative matrix factorization to combine communities and their evolutions in a unified way, which is different from traditional methods in that two-stage techniques are handled separately. However, the FacetNet method requires prior knowledge and must specify the number of communities. Kim and Han [24] introduced an evolutionary clustering method based on particles and density (PDEM). This method treats the network as a collection of tiny particles and divides high-quality communities through cost-embedding technology and modularity optimization. Folino and Pizzuti [25] presented a multiobjective genetic algorithm (DYNMOGA) based on an evolutionary clustering framework to identify the community structure in dynamic networks. DYNMOGA maximizes the cost of the current snapshot by optimizing the modularity to achieve good community partitioning and minimizes timing costs by optimizing the normalized mutual information (NMI). DYNMOGA can achieve a better quality of community discovery, but one of its disadvantages is high time complexity.

**Optimization-based Methods.** Optimization-based approaches handle the community detection by transforming it into an optimization problem, and the optimal solution is obtained by optimizing a predefined objective function. However, this is often an NP-hard problem. Tantipathananandh et al. [26] proposed a framework for community detection in dynamic social networks. In each time step, the social interaction of individuals or organizations is observed in the form of subgraphs. It uses combinatorial optimization algorithms based on dynamic programming, exhaustive search, maximum matching and greedy heuristics to approximate the community structure in the network. One disadvantage of this algorithm is the high time complexity. Niu et al. provided L-DMGA [27], which is a multiobjective optimization genetic algorithm and is based on label propagation dynamics. L-DMGA regards community detection in dynamic networks as a multiobjective optimization problem. Quick Community Adaptation (QCA) [28] is a modularity-based optimization method, and it is also an incremental dynamic community discovery method. QCA uses the Louvain method to perform initial community partitioning and then processes the changed nodes and edges using the objective function in the subsequent snapshot networks. One advantage of the QCA approach is the low time complexity due to the incremental technology it employs. Agarwal et al. [29] introduced a community detection method (DyPerm) based on maximizing permanence in dynamic networks. One disadvantage of the DyPerm method is that it needs to have the initial community structure of the network, but this is unknown in many real networks.

**Other Methods** Xie et al. [15] developed an incremental community detection method (LabelRankT) based on label propagation in dynamic networks. An advantage of this method is its low time complexity (O(m)), which is linear with the edges *m*. However, the LabelRankT algorithm has several parameters that are difficult to set [30]. Xin et al. [31] presented an adaptive random walk sampling method (ARWS), which only updates the adjacent nodes of nodes affected by dynamic events. ARWS is used to detect overlapping community structures in dynamic networks. Quiles et al. [32] introduced a dynamic community detection method based on particle dynamics, which regards the nodes of the network as particles, and employs the diffusive equations of motion and the topological properties of the network to construct a dynamic model of the interaction between particles for clustering. This method needs to set the relative strength parameters of attraction between particles.

In summary, many types of dynamic community detection algorithms have been proposed from different perspectives. Each algorithm has its own advantages and disadvantages, such as parameter setting, high time complexity, and detection accuracy decreasing with time slice. Therefore, efficient community detection in dynamic networks is still a challenging task.

## 3. Methods

### 3.1. Preliminaries

Dynamic community detection is the process of partitioning communities in each snapshot of a continuously changing network. We model a dynamic network as an undirected graph $G_{t}=\left(V_{t},E_{t}\right)$, where *t* represents the time step, $V_{t}$ is the set of nodes, and $E_{t}$ is the set of edges. The definition of community detection in dynamic networks is depicted as follows: Let $DG=\{G_{1},G_{2},\ldots ,G_{T}\}$ be a given dynamic network, where *T* denotes time steps (1≤t≤T). Let $P_{t}=\left\{C_{1},C_{2},\ldots ,C_{k}\right\}$ represent the result of community partition in the snapshot network at time step *t*. The purpose of dynamic community detection is to find an optimal partition $DP=\{P_{1},P_{2},\ldots ,P_{T}\}$. Before elaborating the proposed framework for dynamic community detection, we introduce some basic definitions, which will be used in the following sections.

**Definition 1.** 
*(Jaccard similarity coefficient [33]) Let $G_{t}=\left(V_{t},E_{t}\right)$ be an undirected network at time steps t. The Jaccard similarity coefficient of two nodes v and u is defined as follows:*
(1)$$JS_{vu}=\frac{|\tau (v)⋂\tau (u)|}{|\tau (v)⋃\tau (u)|}$$
*where u∈V, v∈V, τ(u)=N(u)∪{u}, and N(u) is the set of adjacent nodes of node u.*


In real life, social networks often include strong and weak relationships, which play a significant role in information propagation and community formation. To describe this relationship, we use contact strength to represent the degree of tightness between nodes in a given network. Here, we use triangles to formalize the definition of the contact strength because the triangle structure can better characterize the tightness of the vertices.

**Definition 2.** 
*(Contact strength) Let $G_{t}=\left(V_{t},E_{t}\right)$ be an undirected network at time step t, and the contact strength of vertex v on vertex u is defined as follows:*
(2)$$CS_{uv}=\frac{\left|N(u)⋂N(v)\right|}{T_{u}}$$
*where $T_{u}$ denotes the number of triangles for vertex u, and the intersection between N(u) and N(v) represents the amount of triangles shared by two nodes u and v.*


Here, $CS_{uv}$ is an asymmetric function, in other words, the values of $CS_{uv}$ and $CS_{vu}$ may not be equal. For example, everyone has their own circle of friends, and the contact strength between two people is likely to be unequal.

In the real world, the more friends a person has, the more resources he has, so he may obtain more information. To describe the information of nodes in the network, we use the degree of nodes to characterize the initial information of the nodes.

**Definition 3.** 
*(Information) Let $G_{t}=\left(V_{t},E_{t}\right)$ be an undirected network at time step t, and the information of vertex u is defined as follows:*
(3)$$I_{u}=\frac{D_{u}}{D_{max}}$$
*where $D_{u}$ represents the degree of vertex u, and $D_{max}$ denotes the maximum degree of the network $G_{t}$.*


### 3.2. Information Dynamic Model

To reveal the communities in the dynamic networks, we begin to construct the information dynamic model, which includes three parts: information propagation volume, information loss, and propagation model. For a more detailed introduction, please refer to our work in another literature [34].

**Propagation Volume.** According to the research on the patterns of information diffusion among people in the real world, we assume that everyone can acquire information from their neighbors and propagate information to them. The diffusion of information is greatly dependent on the local topology of a network, such as the degree of a node, similarities and connection strengths of nodes. For example, people prefer to choose to communicate with people with whom they have closer links or common interests. To describe the propagation of information in a more realistic way, we employ the node similarity, connection strength, and information difference to model the amount of information diffusion. Formally, let $I_{u\to v}$ represent the information that a node *v* obtains from its neighbor *u*, which is defined as follows:(4)$$I_{u\to v}=f\left(I_{u}-I_{v}\right)JS_{uv}CS_{uv}$$
where $JS_{uv}$ denotes the Jaccard similarity coefficient between node *u* and node *v*, and $CS_{uv}$ represents the contact strength of node *v* on node *u*. The coupling function f(·) denotes the information that can be disseminated from the *u* to *v*, which is defined as follows:(5)$$f\left(I_{u}-I_{v}\right)=\left\{\begin{array}{ccc} e^{\left(I_{u}-I_{v}\right)}-1 & \; & I_{u}-I_{v}\geq 0\\ 0 & \; & I_{u}-I_{v}<0. \end{array}\right.$$
We can see that the nodes with a large information volume are more likely to diffuse and affect the nodes with a small information volume. When the information of $I_{u}$ and $I_{v}$ are close to equal, the amount of information passed between them tends to zero.

The loss of information may occur during the information propagation process in the real world. For example, if the information disseminated is familiar or attractive to us, we can understand and spread it more easily. By contrast, owing to environmental factors, people may misunderstand, ignore, or even lose information. To describe the loss of information in a more realistic and accurate manner, we employ the topological features and information volume for its characterization. Formally, we define the loss of information as follows:

**Information Loss.** In the real world, because of the influence of complex environments, loss may occur during information dissemination. As an example, if people are familiar or interested in the information diffused, they may understand and propagate it more easily. Conversely, we may ignore, misunderstand or even lose information. To reflect the loss of information in a more realistic and accurate way, we use the volume of information and the topological features for its characterization. Let $I_{(u\to v)\_cost}$ denote the loss of information, which is defined as follows:(6)$$I_{(u\to v)\_cost}=\frac{Avg_{s(v)}}{Avg_{d(v)}}f\left(I_{u}-I_{v}\right)*\left(1-JS_{uv}\right)$$
where the first item of the formula characterizes the local topological feature, which consists of the local average similarity and local average degree. $I_{(u\to v)\_cost}$ is positively correlated with coupling function f(·) and negatively correlated with $JS_{uv}$. Therefore, the larger the information volume to be diffused, the greater the information loss is, and the more similar the communication objects are, the smaller the information loss is.

**Information Propagation.** Finally, by considering the information volume and the information loss described above, the information dynamics of a node *v* over time is provided by the following:(7)$$I_{v(t+1)}=I_{v(t)}+\sum_{u\in N(v)}\left(I_{u\to v}-I_{(u\to v)\_cost}\right)$$
where $I_{v(t)}$ represents the information of node *v* at time step *t*, and the second term of this formula denotes the information that is acquired from its neighbors. As we can see, the information of node *v* at time step t+1 includes the information at time step *t* and the information obtained from its neighbors at time step t+1. With the evolution of time, the information propagated tends to zero. Eventually, the information in the network will reach equilibrium, and the communities can be uncovered naturally.

### 3.3. Dynamic Community Detection Framework

Building upon the proposed information dynamics model, we construct a dynamic community detection framework. Figure 2 illustrates the rationale of dynamic community detection based on information dynamics, which consists of two parts: initial community detection and incremental community detection. First, the initial communities $C_{0}$ are obtained by using the information dynamics model at time slice $T_{0}$, and then the dynamic community detection at time slice $T_{t}$ is performed. Next, we introduce the specific process of dynamic community detection based on information dynamics.

**Initial Community Detection.** We employ the information dynamics model to detect the community structure of the whole network at time slice $T_{0}$.

**Incremental Community Detection.** After the initial community detection using the information dynamics model, an incremental community discovery method is adopted for the subsequent snapshot networks. The process of incremental community detection mainly involves the following steps:(1)**Extract Changed Subgraph $\Delta G_{t}$.** Unlike static networks, the structure of dynamic networks is evolving over time. Nodes and edges in a dynamic network may occur or disappear over time, which may lead to constant changes in the community structure of the network. Next, we will analyze network events that may cause changes in community structure, including adding nodes, deleting nodes, adding edges, and deleting edges.
(a)**Add Nodes.** Adding nodes refers to the new nodes added to the current time slice network $G_{t}$ compared with previous time slice network $G_{t-1}$. Let AN denotes the set of added nodes, which is defined as follows:
(8)$$AN=\{v|v\in V_{t},v\notin V_{t-1}\}$$
where $V_{t}$ and $V_{t-1}$ represent the set of nodes in the networks $G_{t}$ and $G_{t-1}$, respectively. The added nodes can be calculated by solving the difference set of these two sets. What is the impact of adding a new node to the community structure in the network? When the added node is within one community, the original community structure will not change [35,36]. As shown in Figure 3b, adding a node inside one community increases the connection density of the community, and the number of communities has not changed. Therefore, it is only necessary to divide the added nodes into the current community.Conversely, when the added node is not inside the community, it may cause a change in the community structure, as shown in Figure 3c. In this case, it is necessary to record the added nodes and the connected communities and add them to the subgraph $\Delta G_{t}$.(b)**Delete Nodes.** The deleted node refers to a node that is removed in the current time slice network $G_{t}$ compared with the previous time slice network $G_{t-1}$. Let AN represent the set of deleted nodes, which is given by the following:
(9)$$DN=\{v|v\notin V_{t},v\in V_{t-1}\}.$$
The deleted nodes can be computed by solving the difference set between sets $V_{t}$ and $V_{t-1}$. Figure 4 shows one node deletions within the community and between communities, respectively. We can observe that the deletion of a node within one community or between the communities has caused changes in the structure of the community. Therefore, when deleting a node, we need to add the deleted node and the connected communities to the subgraph $\Delta G_{t}$.(c)**Add Edges.** Similarly, the added edges correspond to the new edges in the current time slice network $G_{t}$ compared to the previous time slice network $G_{t-1}$. Formally, we define the added edges as follows:
(10)$$AE=\{e|e\in E_{t},e\notin E_{t-1}\}.$$
where $E_{t}$ and $E_{t-1}$ represent the set of edges in the networks $G_{t}$ and $G_{t-1}$, respectively. Figure 5b shows that adding an edge within the community enhances the density of the edges inside it and does not change the structure of the community. Therefore, it is not necessary to deal with the new edges added. However, the addition of edges between communities may lead to changes in community structure. Figure 5c illustrates that the added edges between communities have led to the merger of this two communities into one community.(d)**Delete Edges.** The deleted edge refers to the edge removed in the current time slice network $G_{t}$ compared to the previous time slice network $G_{t-1}$. Let DE denote the set of deleted edges, which is defined as follows:
(11)$$DE=\{e|e\notin E_{t},e\in E_{t-1}\}.$$
Deleting edges within one community may lead to changes in the community structure. As shown in Figure 6b, deleting the inner edge causes the community to split. Therefore, we need to add the current edge and the community involved to the subgraph $\Delta G_{t}$. In contrast, deleting the links between the communities weakens the connection between them, which does not cause changes of the original communities. Figure 6c shows an edge deleted between communities, and the communities have not changed. Therefore, there is no need to process the deleted edges and related communities.(2)**Calculate Changed Communities $\Delta C_{t}$.** After obtaining the subgraph $\Delta G_{t}$ that may change, we need to redetect the communities in the subgraph. Here, we employ the information dynamics to discover the subgraph $\Delta G_{t}$ incrementally and obtain the corresponding community structure $\Delta C_{t}$ (cf. Algorithm 3)(3)**Compute Unchanged Communities $C_{t-1}^{{}^{\prime }}$.** Based on the acquired networks $G_{t-1}$ and $\Delta G_{t}$, we can calculate all the communities and the communities that may change at the t−1 time slice. Therefore, the unchanged communities can be obtained by calculating the difference set of the two sets.(4)**Compute Communities $C_{t}$.** Communities in the network at time slice *t* are composed of the unchanged communities at the previous time slice t−1 and the changed communities at the time slice *t*. Let $C_{t}$ denote the communities of network $G_{t}$ at time slice *t*, which is given as follows:
(12)$$C_{t}=C_{t-1}^{{}^{\prime }}+\Delta C_{t}$$
where $C_{t-1}^{{}^{\prime }}$ represents the unchanged communities at the previous time slice t−1, and $\Delta C_{t}$ denotes the changed communities at the time slice *t*.

Unlike the general real-time incremental algorithm, the incremental method based on information dynamics proposed in this paper adopts batch processing instead of single event processing. The advantage of this method is that it can improve the detection efficiency of the community. In contrast, when the events are handled one by one, the difference in the order of event processing may result in different detection results, and the detection efficiency of the community is also affected [16].

### 3.4. Dynamic Community Detection Algorithm

In this section, we introduce the dynamic community detection algorithm based on information dynamics (DCDID).
(1)**Community Detection based on Information Dynamics**. Based on the information dynamics models, we identify the community structure by simulating the interaction of information on the network, which mainly involves several steps. In the beginning, each node is provided initial information in light of the local topology features (cf. Equation Equation 3). Then, the information diffuses in the network and every node is constantly interacting with neighbor nodes. The exchange of information between nodes in the same community is more frequent than that in different communities. At each step, every node updates its information based on the information dynamics models (cf. Equation Equation 7). As time evolves, the exchange of information between nodes tends to zero, and the information dynamics of each node in the network reaches the convergent state. Finally, the amount of information for each node in the same community is basically the same, and the information on each node in different communities is different. Therefore, we can naturally uncover the communities by considering the amount of information for each node.(2)**Dynamic Community Detection**. DCDID mainly consists of the three steps: initial community structure detection, calculation of subgraphs that have changed, and incremental community identification.
(a)**Initial Community Structure Detection**. The initial community structure is the community partition of the network at time slice $T_{0}$. There is no prior information about community structure in the initial time slice, so it is necessary to perform community detection on the entire network. We use the community detection based on information dynamics (CDID) to identify the community structure of the initial network at time slice $T_{0}$. The CDID algorithm is given in the appendices (Algorithm 2 in Appendix A).(b)**Changed Subgraphs**. Considering the operations that may cause changes in the community structure, we divide the events that change the network into four categories: adding nodes, deleting nodes, adding edges, and deleting edges. Algorithm 3–6 in Appendix A show the specific process, and each type of event returns a subgraph that may change.(c)**Incremental Community Identification**. At present, most incremental dynamic community detection methods adopt the fine-grained processing method, which processes an event when an event is generated. For example, when a node is added to the network, the node is detected. The advantage of this design is that the processing of events is takes place in real time, but the disadvantage is that it increases the computational complexity. Here, we employ a batch-based incremental community detection method. Based on the obtained subgraphs that may change, we employ the information dynamics model to incrementally detect the communities. The DCDID algorithm is given in Algorithm 1.

**Algorithm 1** DCDID**Input:** $DG=\{G_{0},G_{1},\ldots ,G_{k}\}$**Output:** $DC=\{C_{0},C_{1},\ldots ,C_{k}\}$ 1: //Initial community detection 2: $C_{0}=CDID\left(G_{0}\right)$ 3: //Incremental community detection 4: **for**
t=1 to *k*
**do** 5:  compute AN,DN,AE,DE using Equation (Equation 8)–(Equation 11) 6:  $\Delta G_{t}\leftarrow Add\_nodes\left(AN,G_{t},C_{t-1}\right)$ 7:  $\Delta G_{t}\leftarrow Del\_nodes\left(DN,G_{t-1},C_{t-1}\right)$ 8:  $\Delta G_{t}\leftarrow Add\_edges\left(AE,C_{t-1}\right)$ 9:  $\Delta G_{t}\leftarrow Del\_edges\left(DE,C_{t-1}\right)$ 10:  compute the unchanged communities $C_{t-1}^{{}^{\prime }}$ 11:  $\Delta C_{t}\leftarrow CDID\left(\Delta G_{t}\right)$ 12:  compute $C_{t}$ using Equation (Equation 12) 13:  **end for**

### 3.5. Complexity Analysis

The time complexity of the DCDID algorithm is mainly composed of two parts: one is the time complexity of the initial community partition, and the other is the time complexity of the incremental community detection.
(1)**Initial Community Partition**. In the beginning, community detection is required for the entire network, so the time complexity is the time taken by the CDID algorithm. The CDID algorithm consists of three steps: information initialization, information dynamic interaction, and community partition. In the first step, CDID needs to compute the initial information, Jaccard similarity coefficient and contact strength. Thus, the time complexity of information initialization is O(k·n), where *k* denotes the average degree, and *n* represents the number of nodes in the network. In the second step, the time complexity of information interaction is O(L·n·k) due to the local interaction strategy, where *L* denotes the number of iterations. It is typically between 20 and 100. In the third step, the time complexity is O(k·n) because of two loops for finding communities. Thus, the time complexity for the initial community partition is O(L·n·k).(2)**Incremental Community Detection**. The incremental community detection mainly includes the calculation of the changed subgraph and the incremental partition communities. The calculation of the subgraph consists of adding nodes, deleting nodes, adding edges, and deleting edges. The time complexity of adding nodes is $O(|\Delta V_{t}|\cdot k_{t})$, where $\Delta V_{t}$ is the nodes set added and $k_{t}$ is the average degree at time slice *t*. Similarly, the time complexity of deleting nodes is $O(|\Delta V_{t}|\cdot k_{t})$. The time complexity of adding edges and deleting edges is $O(|\Delta E_{t}|)$ because of only one loop. The next step is to detect the communities of the subgraph ΔG that have changed, and the time complexity is $O(L\cdot |\Delta V_{t}|\cdot k_{t})$. Thus, the time complexity for incremental community detection is $O(|\Delta E_{t}|+L\cdot |\Delta V_{t}|\cdot k_{t})$.

In summary, the time complexity of the DCDID algorithm is divided into the complexity O(L·n·k) at the time slice $T_{0}$ and the time complexity $O(|\Delta E_{t}|+L\cdot |\Delta V_{t}|\cdot k_{t})$ at the time slice $T_{t}$. In general, $|\Delta V_{t}|$,$|\Delta E_{t}|$ and $k_{t}$ are very small, so the time complexity at the time slice $T_{t}$ is relatively low. Therefore, the DCDID method can be applied to dynamic community detection of large-scale networks.

## 4. Experiments

In this section, we evaluate our dynamic community detection method DCDID on real-world and synthetic networks to demonstrate its benefits. To extensively research the performance of DCDID, we compare it with several representative dynamic community detection algorithms. Before the empirical comparison, we briefly introduce the comparison algorithms.

**QCA** [28] is a modularity optimization algorithm based on Louvain [9]. It adaptively updates and detects new community structures according to the changes of network structure and network information in the previous time slice.

**FacetNet** [23] is a well-known dynamic community detection algorithm, which uses nonnegative matrix decomposition to analyze the community structure and evolution in dynamic networks and optimizes the quality of detected communities through a loss function. FacetNet requires parameter settings, such as the number of communities.

**DYNMOGA** [25] is a multiobjective optimization genetic algorithm based on evolutionary clustering. It detects the community structure in dynamic networks by optimizing Modularity and NMI. The DYNMOGA algorithm also requires parameter settings.

**DyPerm** [29] is an optimization method based on permanence, which is also an incremental dynamic community detection method. DyPerm needs to specify the actual communities at the beginning.

**InBatch** [16] is an incremental dynamic network community detection algorithm based on the Louvain method. It processes the changed network structure in batches instead of an event-by-event approach.

**LBTR** [14] is an incremental dynamic community detection method based on machine learning. It also uses the Louvain method to obtain the initial community structure, and then implements the machine learning method for classification, prediction and revision.

### 4.1. Data Description

**Synthetic Networks.** Although there are many dynamic networks in the real world, we rarely know the ground truth of communities. Therefore, we constructed many synthetic networks with ground truth communities to evaluate algorithms. To obtain a synthetic network that is more similar to a real-world network, we employ the extended LFR [37] model to generate dynamic networks. The extended LFR benchmark can be easily controlled by several parameters, such as the number of time slices, average degree, community size, mixing parameter, and several events that cause the structure of the network to change. Table 1 shows the detailed descriptions of parameters for this benchmark model.

**Real-world Data Sets.** To evaluate the effect of the dynamic community detection algorithms more comprehensively, we select several different scales of real-world networks with ground truth. Table 2 shows the statistical properties of each dynamic network, where $\bar{N}$ denotes the average number of nodes, $\bar{M}$ represents the average number of edges, $\bar{k}$ is the average degree, $\bar{CC}$ is the average clustering coefficient, and *S* denotes the number of time slices. All these dynamic networks are available at the website of social patterns (http://www.sociopatterns.org/datasets/) and the data repository of citation networks (https://www.aminer.cn/citation). Next, we briefly describe these real-world networks.

**2011 High school dynamic contact networks (HSD11):** This data set contains the time series network of contacts between three classes in a high school in Marseille, France, in December 2011. HSD11 includes 7 time slices, and the network of each time slice contains three communities. In HSD11 dynamic network, one node represents a student, and one edge indicates that there is a connection between the students.

**2012 High school dynamic contact networks (HSD12):** Similar to HSD11, this data set is also a time series network for connections between high school students in Marseille, France. This network was collected in November 2012 and consists of five classes. The other information is consistent with HSD11.

**Primary school contact networks (PS):** This data set contains a time series network of contacts between children and teachers. Each child or teacher represents a node, and the contact between them represents an edge.

**Contact network in a workplace (CW):** This data set is a time series network of contacts between people in an office building in France. The CW dynamic network consists of five departments as real communities and records the contact between people at intervals of 20 seconds.

**Cumulative coauthorship network (CC):** This data set is a collaboration network derived from the citation network. The data set used in this paper has been collated and modified by reference [38]. In this dynamic network, one node represents an author of one paper, and an edge denotes the relationship between the coauthors of a paper.

**Noncumulative coauthorship network (NCC):** This data set is also a collaboration network similar to the CC dynamic network. The CC dynamic network accumulates the changes of each node and edge. In contrast, the NCC dynamic network does not accumulate, i.e., when two authors copublish papers many times, there is still only one link between them.

### 4.2. Evaluation Metrics

At present, the evaluation of the performance of the dynamic community detection is mainly to quantify the goodness of community detection in each time slice, and the evaluation criteria adopted are consistent with community detection in a static network. Here, we employ NMI [39] and ARI [40] metrics to evaluate the performance of the comparison algorithms. NMI and ARI are widely used for quantifying the quality of communities detected by algorithms. Before presenting the experiment, let us briefly describe the evaluation metrics.

Normalized mutual information (NMI) is a well-known metric, which originates from information theory. NMI is widely used to evaluate the result of disjoint community detection, which posits that if two partitions of communities are similar, then little additional information is needed to deduce one division from the other. Formally, the definition of NMI is given as follows:(13)$$NMI\left(A;B\right)=\frac{2I(A;B)}{H(A)+H(B)}$$
where *A* and *B* are the partitions of communities, I(A;B) denotes the mutual information of random variables *A* and *B*, and H(A) represent the entropy of *A*. The value of NMI ranges from 0 to 1, where 0 represents that the detected communities are completely independent of the real communities, whereas 1 denotes a perfect match with the ground truth.

Adjusted rand index (ARI) is another measure to evaluate the similarity between two communities, which is defined as follows:(14)$$ARI=\frac{RI-ExpectedRI}{MaxRI-ExpectedRI}$$
where RI denotes the similarity of two partitions, which includes all pairs of samples. Next, it calculates the numbers of pairs that are assigned to the same or different partitions in the predicted and true partitions [41]. More specifically,
(15)$$\mathrm{ARI}=\frac{\sum_{ij}\left(\begin{array}{c} n_{ij}\\ 2 \end{array}\right)-\left[\sum_{i}\left(\begin{array}{c} a_{i}\\ 2 \end{array}\right)\sum_{j}\left(\begin{array}{c} b_{j}\\ 2 \end{array}\right)\right]/\left(\begin{array}{c} n\\ 2 \end{array}\right)}{\frac{1}{2}\left[\sum_{i}\left(\begin{array}{c} a_{i}\\ 2 \end{array}\right)+\sum_{j}\left(\begin{array}{c} b_{j}\\ 2 \end{array}\right)\right]-\left[\sum_{i}\left(\begin{array}{c} a_{i}\\ 2 \end{array}\right)\sum_{j}\left(\begin{array}{c} b_{j}\\ 2 \end{array}\right)\right]/\left(\begin{array}{c} n\\ 2 \end{array}\right)}$$
where $a_{i}$, $b_{j}$ and $n_{ij}$ are values from the contingency table. For details, please refer to [42].

### 4.3. Performance Evaluation

In these experiments, we employed the parameters of comparison algorithms to the default values suggested by the authors. The DyPerm is available at GitHub (https://github.com/ayush14029/Dyperm-Code). The codes of LBTR and InBatch are provided by Jiaxing Shang [14]. QCA, FacetNet, DYNMOGA, and DCDID are available at GitHub (https://github.com/sunwww168/DCDID). For each dynamic network, the average result of each algorithm was obtained by averaging 20 independent runs. All experiments were performed on a desktop computer with a 3.3 GHz CPU of an Intel Core i5 and 16.0 GB RAM.

**Evaluation on Synthetic Networks.** The changes of the community structure in dynamic networks mainly include: the nodes’ switching community membership in each time slice, the birth and death of communities, the growth and contraction of communities, and the merger and split of communities. To evaluate the performance of the proposed dynamic community detection algorithm, we employ the dynamic LFR benchmark model to generate multiple synthetic networks with different characteristics. To cover several states of dynamic network community structure changes, we evaluate the comparison algorithms from four aspects: node switch, community birth and death, community expansion and contraction, and community merger and split. Without loss of generality, the common parameters of these states are set as follows: the time slices s=20, the number of nodes in each time slices n=1000, the average degree k=[5−25], and the max degree maxk=[20−50]. The DyPerm algorithm does not take part in the comparison of the initial time slice because of it uses the ground truth as the initial community information.
(1)**Node Switch**. Node switch refers to the transition of a node from one community to another in different time slices in a dynamic network. The parameter *p* represents the probability that a node switches community membership in different time slices. We varied the value of *p* from 0.1 to 0.8 and fixed the parameters k=10, maxk=20, and μ=0.1. Figure 7 shows the performance of comparison algorithms with different *p* on NMI and ARI metrics. Because of the limitation of space, we only display the results of community detection when p is 0.1, 0.4 and 0.8. In terms of the NMI metric, DCDID and DYNMOGA methods acquired the best effects, and the values of NMI obtained on each time slice were approximately 0.95. It demonstrates that the performances of these two methods are relatively stable over time. FacetNet also performed well, and the NMI values achieved reached to 0.9. However, the FacetNet algorithm needs to specify the number of communities in the network, which is often unknown in the real world. DyPerm also had a stable performance, and the values of NMI were basically maintained at approximately 0.8. QCA, InBatch, and LBTR methods initially implemented very high NMI values because these algorithms use the Louvain method to detect the community structure of the initial time slice. However, the performances of these algorithms gradually declined over time. In particular, the NMI values obtained by the InBatch algorithm were close to 0.2 when *p* was larger than 0.4. In terms of the ARI metric, DCDID obtained the best effect, which achieved the highest ARI value among these algorithms. DYNMOGA and FacetNet methods acquired better results than the other comparison algorithms. Although DyPerm attained good NMI values, its ARI values were very low. Because the granularity on the community partition of the DyPerm algorithm became increasingly fine with the increase of time slices, the number of communities it partitions was usually several dozen times the number of real communities. The ARI values of QCA, InBatch and LBTR algorithms also decreased with the increase of time slices. In particular, the values of ARI were close to zero when *p* was larger than 0.4 and the time slice was greater than 10.(2)**Community Birth and Death**. To investigate the effects of comparison algorithms on the birth and death event of communities, we fixed parameters k=10, maxk=20, μ=0.1, and p=0.1, and varied the number of birth and death communities to generate dynamic networks. Because the dynamic LFR model cannot generate a dynamic network when the number of birth and death communities both reach 16, we changed the number of birth communities from 2 to 16 and varied the number of death communities from 2 to 8. Figure 8 reveals the performance of each algorithm on NMI and ARI metrics with different numbers of birth and death communities. In terms of the NMI metric, DCDID and DYNMOGA were very stable, and they obtained higher values than the other algorithms. FacetNet cannot run on this dynamic network because the number of communities in each time slice is constantly changing over time. DyPerm also performed well with the NMI values stable at approximately 0.8, but the ARI values were very low. In particular, its values were close to zero when the number of birth and death communities were greater than 8 and the time slice was greater than 12. QCA and LBTR achieved acceptable results, and they also obtained an NMI value of 0.5 when the number of birth communities reached 16. By contrast, InBatch did not perform well in this group of experiments and obtained the lowest values of NMI and ARI.(3)**Community Expansion and Contraction**. To further evaluate the performance of each algorithm on the expansion and contraction event of communities, we fixed parameters k=10, maxk=20, μ=0.1, and p=0.1, and varied the number of expansion and contraction communities from 5 to 40. As shown in Figure 9, DCDID and DYNMOGA acquired the best quality of community detection. Their NMI values were stable at approximately 0.95, and ARI values were stable between 0.8 and 0.9. FacetNet also yielded good results, and its NMI values were stable at approximately 0.84. Although the DyPerm method achieved stable NMI values at approximately 0.8, the ARI values were relatively low, indicating that the quality of its community detection was not ideal. The effectiveness of QCA, InBatch and LBTR methods decreased gradually over time. It explains that these algorithms have higher cumulative errors.(4)**Community Merger and Split.**Figure 10 describes the effectiveness of comparison algorithms when the number of merger and split communities varied from 5 to 40. We can observe that DCDID performed best in these comparison algorithms, and the NMI value of each time slice was stable at approximately 0.96 when the number of merger and split communities was less than 20. DYNMOGA also performed well, which achieved better results of community detection than other comparison algorithms. As shown in Figure 10c, the results of DCDID, DYNMOGA, FacetNet and DyPerm algorithms fluctuated greatly when the number of communities merged and split in dynamic networks reaches 40, i.e., almost all communities in the network have changed. We can see that the NMI and ARI values obtained were lower in the seventh time slice, but the quality of the community detection of these four algorithms was still better than other comparison algorithms. The performance of QCA, InBatch and LBTR algorithms also decreased gradually over time. Interestingly, the NMI values of InBatch had a higher improvement when the number of communities merged and split increased to 40 in the 13th time slice (Figure 10c). We analyzed the network structure of the current time slice and found that the community structure under this time slice was relatively clear, and there were fewer links between the communities. This may also be the reason the NMI values of the DCDID, DYNMOGA and FacetNet algorithms had a peak in this time slice.

**Real-world Networks.** To further compare DCDID and the other algorithms, we tested their performances on several real-world dynamic networks with ground truth. The NMI and ARI indexes are employed to evaluate the efficiency of these algorithms. Figure 11 and Figure 12 show the evaluation results of comparison algorithms on real-world dynamic networks. In the CC and NCC dynamic networks, DCDID and DyPerm achieved the best quality of community detection and obtained NMI values of approximately 0.5. In terms of the ARI metric, most of the comparison algorithms did not perform ideally on these two networks, which may be related to their low-average degree. As shown in Table 2, the average degree of each time slice is 4.3. However, the DCDID method acquired the highest value compared with the other algorithms. It indicates that DCDID can still achieve better quality of community detection in the dynamic networks with a low-average degree. DYNMOGA and FacetNet methods cannot run on these two dynamic networks because the number of nodes in them have reached 100,000. QCA obtained acceptable performance on these two real networks, and the results of community detection are better than that of the InBatch and LBTR methods.

On CW, HSD11 and HSD12 dynamic networks, the InBatch algorithm performed very well, where it obtained the highest NMI and ARI values. The DCDID algorithm also achieved good quality of community detection on these three networks, and the results are better than those of the DYNMOGA, QCA and DyPerm algorithms. In the HSD11 and HSD12 dynamic networks, LBTR yielded stable results, similar to DCDID. On the PS dynamic network, DCDID and LBTR obtained the best quality of community detection, and the NMI and ARI values obtained were approximately 0.9 and 0.8, respectively. The QCA method also produced acceptable results on this network. The DyPerm algorithm performance was not ideal, and it failed to identify the communities in CW, HSD11, HSD12 and PS networks.

In summary, many experiments on synthetic dynamic networks demonstrated that the DCDID method achieved good quality of community detection under different network events. The evaluation on the real-world dynamic network shows that DCDID cannot only achieve good community recognition on low-average degree networks but also achieve good community detection performance on other dynamic networks with different characteristics. The DyPerm algorithm performed well on most synthetic networks, but its community divisions were too fine-grained, which resulted in too low ARI values. In addition, this algorithm uses ground truth community information as the initial information; however, this is unknown in most real dynamic networks. DYNMOGA and FacetNet methods performed well on generation networks, but their spatial complexity is high, and they cannot be run on real networks CC and NCC. In addition, the FacetNet method requires prior knowledge and must set the number of communities. QCA, InBatch and LBTR had significant cumulative errors on the synthetic dynamic networks, and their performance degraded rapidly over time. However, InBatch and LBTR methods have achieved good quality of community detection in real-world dynamic networks. In particular, the InBatch method achieved the best quality of community detection on CW, HSD11 and HSD12 networks.

### 4.4. Runtime

To evaluate the scalability of DCDID at different network scales, we employed the dynamic LFR model to generate dynamic networks with different sizes. We varied the number of node ranges from 1K to 100M and fixed the parameters at k=[10−20], maxk=[20−50], p=0.1, μ=0.1, and s=5. Figure 13 reveals the overall running time of the comparison algorithms in five time slices of the dynamic networks. We observed that the DyPerm algorithm used the highest runtime, and the runtime was more than ten days when the number of nodes reaches 5M. The DCDID algorithm was faster than the DyPerm, FaceNet and DYNMOGA methods, and this advantage was obvious as the size of the network increased. This difference was mainly due to the low time complexity of $O(|\Delta E_{t}|+L\cdot |\Delta V_{t}|\cdot k_{t})$, where $|\Delta V_{t}|$,$|\Delta E_{t}|$ and $k_{t}$ are very small. Therefore, the DCDID algorithm can handle large-scale dynamic networks. When the number of nodes reaches 5M, FaceNet and DYNMOGA methods indicated that there was insufficient memory space to run. Although DCDID was slower than QCA, InBatch and LBTR, its quality of community detection was higher than these three algorithms.

## 5. Conclusions

In this paper, we proposed an information dynamics-based dynamic community detection framework, which uses a batch processing technique to incrementally uncover community structures in dynamic networks. In addition, we designed a new algorithm called DCDID and used the information dynamics model to simulate the exchange of information between nodes in each time slice of dynamic networks. DCDID provides an intuitive and topologically driven manner for community detection in dynamic networks. To validate the performance of DCDID, we compared it with six representative dynamic community detection algorithms on synthetic and real-world datasets. Experiments demonstrate that DCDID performed well at dynamic community detection and obtained better results than the representative methods compared in the evaluation.

## Figures and Tables

**Figure 1 entropy-22-00425-f001:**
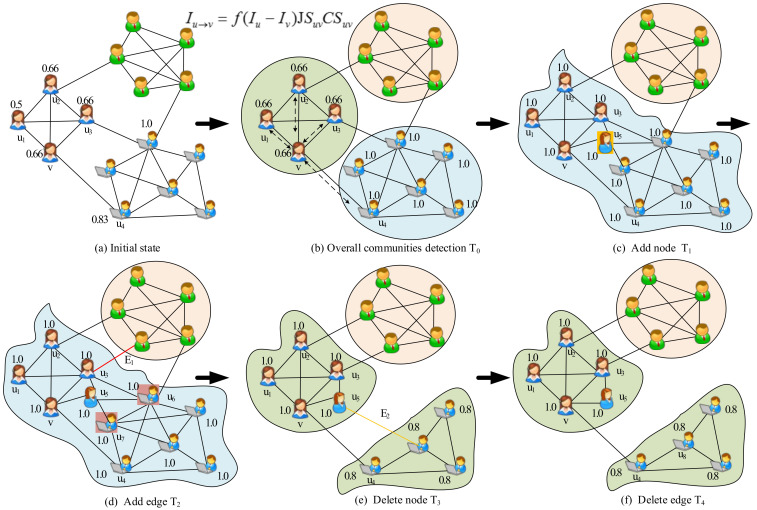
Illustration of dynamic community detection based on information dynamics. (**a**) In the initial state, each person possesses his/her own initial information. (**b**) The community detection based on information dynamics in the overall network at the time slice $T_{0}$. (**c**) The addition of node $U_{5}$ resulted in the merging of the two communities. (**d**) Adding the edge $E_{1}$ did not change the original community partition. (**e**) The deletion of nodes $U_{6}$ and $U_{7}$ caused the community to split. (**f**) Deleting the edge $E_{2}$ did not cause the community to change.

**Figure 2 entropy-22-00425-f002:**
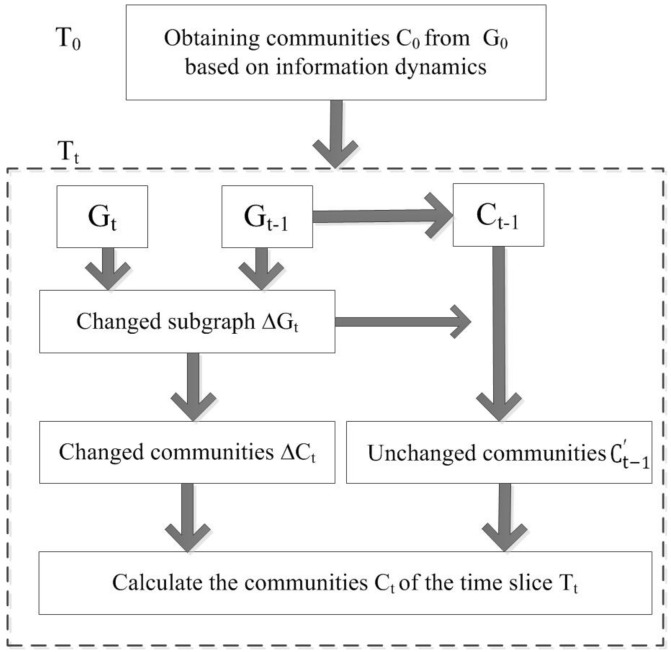
The framework of dynamic community detection based on information dynamics.

**Figure 3 entropy-22-00425-f003:**
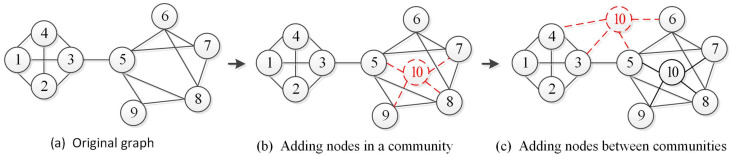
Adding nodes in the network.

**Figure 4 entropy-22-00425-f004:**
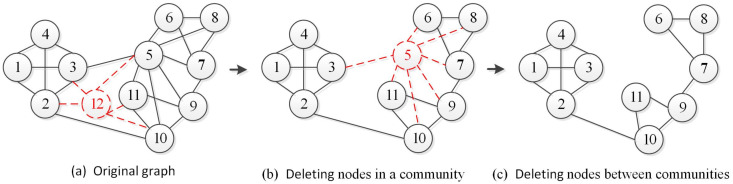
Deleting nodes in the network.

**Figure 5 entropy-22-00425-f005:**
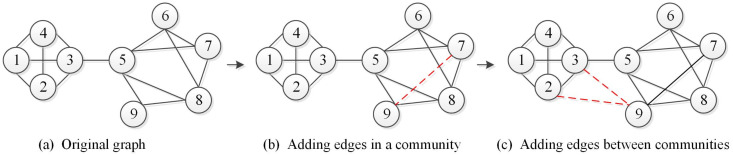
Adding edges in the network.

**Figure 6 entropy-22-00425-f006:**
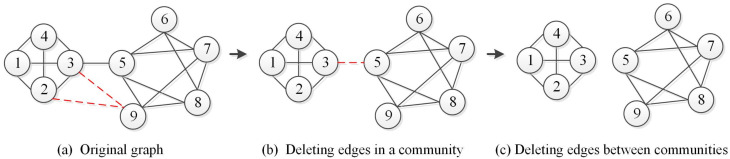
Deleting edges in the network.

**Figure 7 entropy-22-00425-f007:**
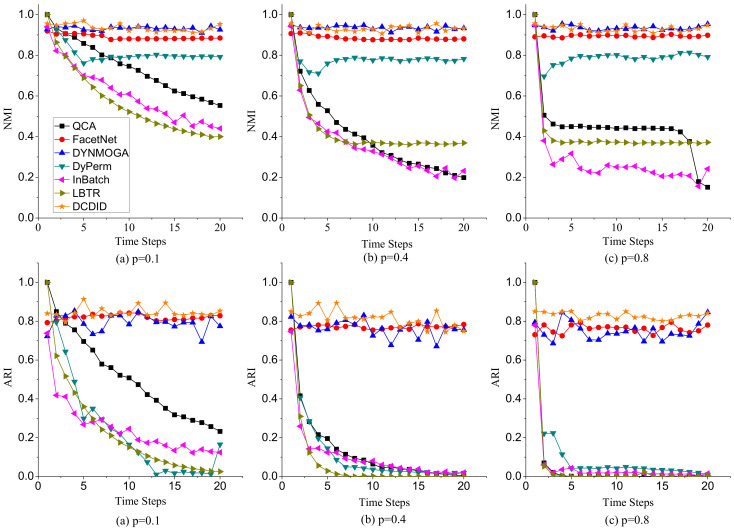
Performances of comparison algorithms on NMI and ARI metrics with different switch probability *p*.

**Figure 8 entropy-22-00425-f008:**
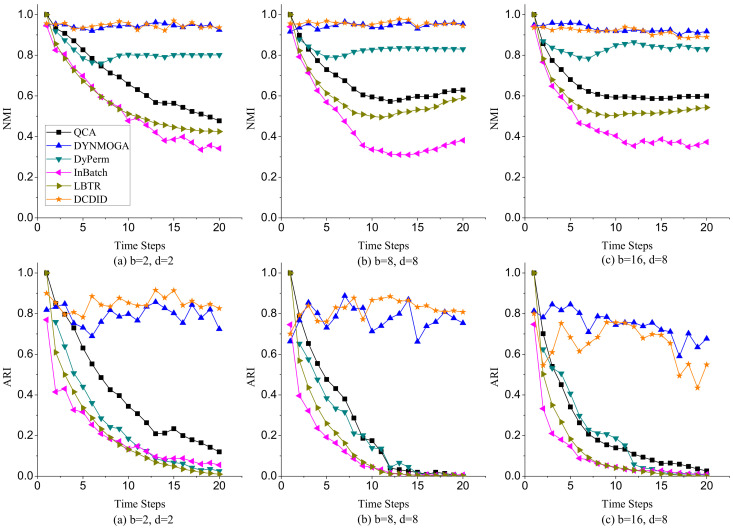
Performances of comparison algorithms on NMI and ARI metrics with different birth and death communities.

**Figure 9 entropy-22-00425-f009:**
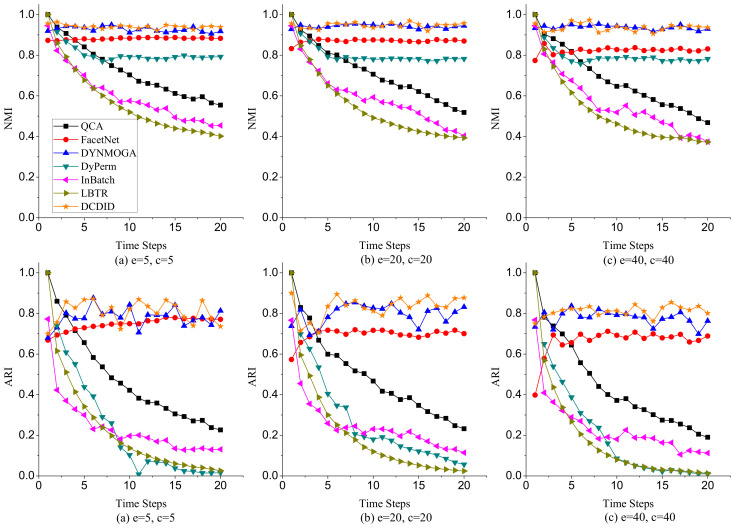
Performances of comparison algorithms on NMI and ARI metrics with different expansion and contraction communities.

**Figure 10 entropy-22-00425-f010:**
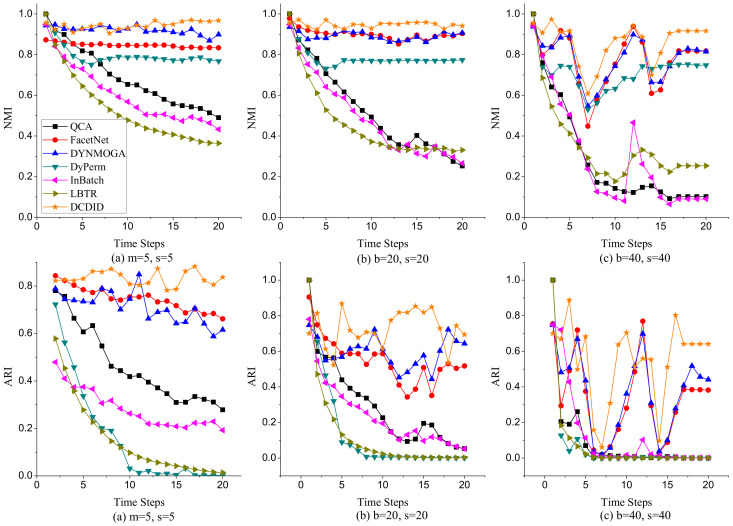
Performances of comparison algorithms on NMI and ARI metrics with different merger and split communities.

**Figure 11 entropy-22-00425-f011:**
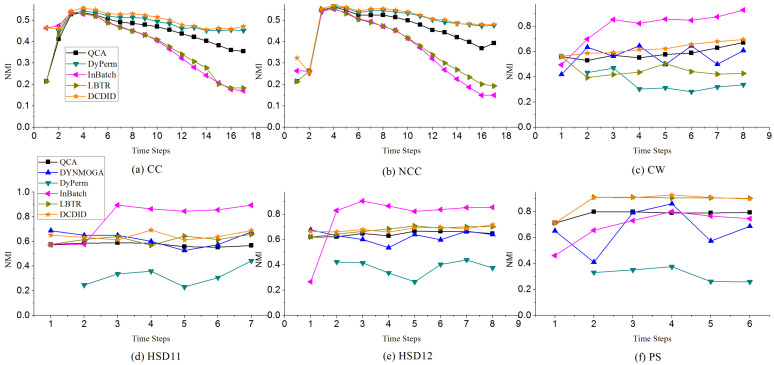
Performances of comparison algorithms on the NMI metric with different merger and split communities.

**Figure 12 entropy-22-00425-f012:**
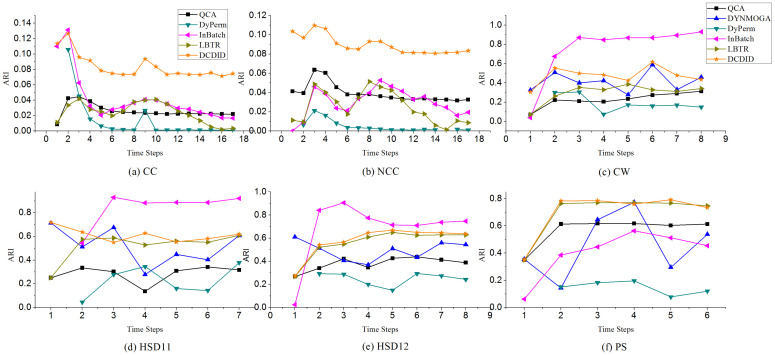
Performances of comparison algorithms on the ARI metric with different merger and split communities.

**Figure 13 entropy-22-00425-f013:**
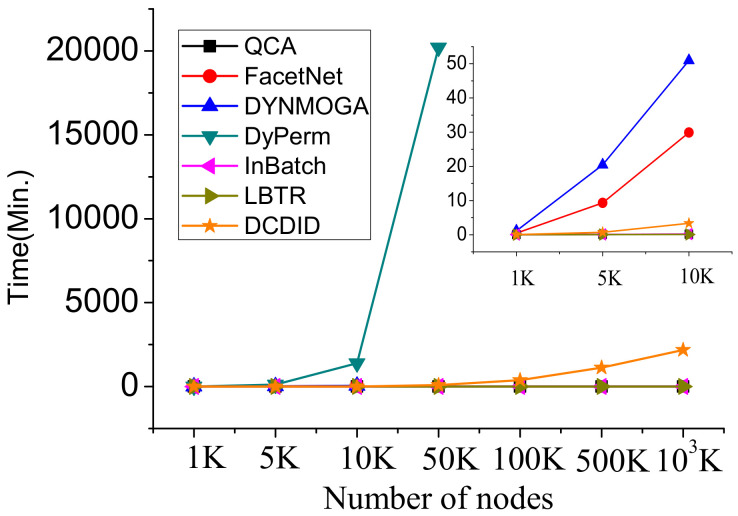
The runtimes of the comparison algorithms on the dynamic LFR benchmark with nodes ranging from 1K to $10^{3}K$.

**Table 1 entropy-22-00425-t001:** Description of the parameters for the dynamic LFR benchmarks.

Symbol	Definition
*n*	number of nodes
*s*	number of time slices
μ	mixing parameter
*k*	average degree of each time slice
$k_{max}$	max degree of each time slice
$C_{min}$	minimum of community sizes for each time slice
$C_{max}$	maximum of community sizes for each time slice
*p*	probability of one node switching community membership between time slices
birth	number of community births for each time slice
death	number of community deaths for each time slice
expand	number of community expansions for each time slice
contract	number of contractions for each time slice
merge	number of community mergers for each time slice
split	number of community splits for each time slice

**Table 2 entropy-22-00425-t002:** Some statistical properties of several real-world networks.

Dataset	$\bar{N}$	$\bar{M}$	$\bar{k}$	$\bar{\mathrm{CC}}$	*S*
HSD11	123	1271	20	0.51	7
HSD12	175	1629	18	0.43	8
PS	239	6146	50.8	0.52	9
CW	88	537	11.4	0.38	8
CC	107,180	376,567	4.3	0.49	17
NCC	107,166	376,543	4.3	0.49	17

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
