# Peer review of "Identifying Communities in Dynamic Networks Using Information Dynamics"

_entropy, 2020, doi:10.3390/e22040425_

Round 1

Reviewer 1 Report

This work presents a novel approach for solving the dynamic community detection problem based on information dynamics. The paper is well written and the proposal is supported with several computational experiments including the scalability of the method. I recommend to accept this work after considering the following minor issues:
- The citation of references [1-6] in line 18 should be justified. Why citing 6 works and not only 1, for instance, if they are not explained? It seems to be unnecessary auto-citation.
- Again, citation [14-16] in line 50 is not justified
- Replace D_max by D_{max} along the manuscript. Revise other terms as Avg_s and Avg_d, which should be with subscript
- The experimental results show that the proposal might be better but it should be justified with statistical tests, since it is very similar to DYNMOGA in several instances.

Author Response

Dear reviewer:

    We are very grateful to your insightful comments for our manuscript # entropy-746486 entitled “Identifying communities in dynamic networks using information dynamics”. Thank you very much for the constructive suggestions and for providing us with an opportunity to improve it. In the following we give point-by-point responses to your comments. All responses to reviewers and corrections to the manuscript have been marked in blue.

Reviewer 2 Report

The authors introduce a novel approach for community discovery in dynamic networks and test it against state of art competitors both on synthetic and real-world datasets.

The proposed approach is indeed interesting, the modeling choices, motivation, and definition provided are sound. However, a few remarks prevent to accept the paper as it is:

  • [Mandatory] Code Availability and experiments replicability: the authors must provide links to the implementation of the proposed algorithm in a public repository system (GitHub, GitLab...). The same for the implementation of competitor algorithms, tools used to generate synthetic networks and procedures to evaluate the obtained results. Without such information, the contribution cannot be accepted for publication.
  • Figure 1: Communities are identified by information value shared by nodes (note that so far you have not described your algorithm so this is not exactly easy to get at a first view): make this information clear in the caption or change the figure to better reflect it (e.g., use the same visual representation for nodes in the same community).
  • Definition 1 (Jaccard): tau(u) must not contain u itself, otherwise the coefficient will range in [0,1) not in [0,1] as it should.
  • Definition 2 (contact strength): you are assuming an undirected network, however the T_u in the denominator design an asymmetric function. Although this makes sense please clarify such discrepancy in the text.
  • Definition 3 (Information): given the normalization, the distribution of the information will tend to be flattened toward low values for long-tailed degree distributions. You coherently discuss such effect in the following pages, however, could you provide insights on how much the obtained communities are separated in terms of their final information scores?
    It seems to me that for long-tailed degree distribution there is a huge risk of obtaining very small differences among node values (that can be affected among other things by machine precision for floating-point number representation) thus producing non-reliable partitions.
  • Page 10: the acronym CDID is used here for the first time but its meaning specified only later on: remove the last sentence on this page.
  • Page 12: you state that "In general [..] so the time complexity at the time slice T_1 is relatively low". Indeed, that heavily depends on the windowing applied to build the snapshot graph. The more unstable the temporal partition (too wide or too small) the more likely that the changes across subsequent snapshots are significant. You cannot assume that the impact of node/edge addition/removal will be as low as you might expect: that's a problem when using snapshot graphs, you have to rely on some external temporal discretization of the network dynamics.
  • Experiments: I would like to see an experiment - even on synthetic graphs - that shows what's the real approximation that you are obtaining by applying the incremental updates instead of recomputing the partition from scratch using CDID for each network snapshot. I get the reduction of complexity and execution time but it would be nice to understand how much the final results differ from the "stable" ones so to understand the trade-off between approximation and runtime.
  • Synthetic benchmark: you are showing results for extended LFR assuming a mixing coefficient mu=0.1. It would be interesting to have some results for higher values of such a coefficient (in the range [0.1-0.5) so to understand how the methods perform when the community structure is loosely defined. Indeed the proposed method, given the local updates, is well-suited for neatly separated communities (less information spreading from their border): do the shown results hold also for loosely defined ones?

Finally, a recent survey on DCD that the authors could consider for updating references on synthetic models for dynamic graphs and algorithms is [1].

[1] Rossetti, Giulio, and Rémy Cazabet. "Community discovery in dynamic networks: a survey." ACM Computing Surveys (CSUR) 51.2 (2018): 1-37.

Author Response

(The authors gave the same response as above.)
